# Interface Engineering of ZnO-Decorated ZnFe_2_O_4_ for Enhanced CO_2_ Reduction Performance

**DOI:** 10.3390/molecules30193980

**Published:** 2025-10-04

**Authors:** Congyu Cai, Yufeng Sun, Yulan Xiao, Weiye Zheng, Minhui Pan, Weiwei Wang

**Affiliations:** 1School of Life Science and Chemistry, Minnan Science and Technology College, Quanzhou 362332, China; caicongyu@mku.edu.cn (C.C.); panminhui@mku.edu.cn (M.P.); 2School of Humanities, Minnan Science and Technology College, Quanzhou 362332, China; sunyufeng@mku.edu.cn

**Keywords:** ZnO/ZnFe_2_O_4_ nanoparticle, oxygen vacancies, photocatalytic CO_2_ reduction, interface engineering

## Abstract

Photocatalytic conversion of CO_2_ to hydrocarbon fuels offers a promising pathway for sustainable renewable energy production. In this study, a ZnO/ZnFe_2_O_4_ composite featuring a Type-II heterojunction was synthesized through a facile one-step hydrothermal approach, significantly enhancing visible-light-driven CO_2_ reduction activity. The optimized catalyst exhibits CH_4_ and CO production rates that are 3.3 and 4.9 times higher, respectively, than those of pristine ZnFe_2_O_4_ over 6 h. This significant enhancement in photocatalytic performance is attributed to the Type-II band alignment, which not only broadens light absorption but also greatly promotes efficient charge separation. It is corroborated by a series of experimental evidence: a two-fold enhancement in photocurrent response, a 15.1% reduction in PL intensity, decreased electrochemical impedance, and an extended charge carrier lifetime. Furthermore, in situ FTIR spectroscopy confirms that the heterojunction facilitates the formation of key intermediates (specifically *COOH and HCOO^−^). This study highlights the importance of precise interface design based on a Type-II heterojunction in heterostructured composite catalysts and provides mechanistic insights for developing highly efficient CO_2_ photoreduction systems.

## 1. Introduction

The rapid development of global industrialization and urbanization has led to a severe environmental crisis caused by CO_2_, posing significant environmental challenges for researchers. Currently, mitigating greenhouse gases primarily through photocatalysis, electrocatalysis, and carbon capture is anticipated to be a key strategy for addressing this crisis [1,2,3,4]. Numerous semiconductor and photocatalytic materials, such as TiO_2_, ZnO, ZnIn_2_S_4_, CeO_2_, and ZnFe_2_O_4_, have been developed [5,6,7,8,9,10]. Among these, photocatalysis represents a widely used green synthesis method in environmental science. It offers advantages including environmental friendliness, lower production costs, and safety. The synthesized nanoparticles exhibit enhanced biocompatibility and good performance, making them suitable for various applications like wastewater treatment, environmental protection, and other bio-related fields [11,12].

However, utilizing light energy to drive the conversion of CO_2_ into carbon-based fuels is challenging due to the high stability and low reactivity of the CO_2_ molecule, attributed to the high dissociation energy of its C=O bond. The conversion of solar energy through mild light-driven chemical reactions is of profound importance for developing green and sustainable energy [13]. Related studies have demonstrated that ZnFe_2_O_4_ nanoparticles possess strong antioxidant activity and high safety. As a novel narrow-bandgap semiconductor material, nano-sized zinc ferrite has been extensively researched and applied in areas such as magnetism, photocatalysis, and energy storage [14,15,16]. Deng et al. [17] prepared monodisperse ferrite microspheres using a simple one-step solvothermal method. Modulating the hydrophilicity and biocompatibility of these micro/nanomaterials is expected to broaden their applications in advanced magnetic materials, ferromagnetic fluid technology, and biomedicine. Zinc ferrite nanomaterials are synthesized via a variety of methods, including mechanical ball milling, sol-gel, solvothermal synthesis, chemical vapor deposition, and microwave-assisted combustion. The advantages of each method are accompanied by certain trade-offs, such as encompassing elevated impurity levels in the final product, procedures that are both cumbersome and time-consuming, or a dependence on stringent conditions like high temperature and pressure [18]. Therefore, more effective synthesis methods are needed to address these material shortcomings.

Meanwhile, ZnO has a narrow light absorption range, primarily limited to ultraviolet (UV) light, which constitutes only about 4% of solar radiation, thus restricting the full utilization of sunlight. Furthermore, the rapid recombination of photogenerated carriers prevents effective separation of electron–hole pairs and their transfer to the catalyst surface for participation in photocatalytic reactions, leading to reduced photocatalytic activity. Strategies reported to enhance the catalytic activity of ZnO include morphology control, ion doping, semiconductor coupling, and surface noble metal deposition. However, these approaches often suffer from drawbacks such as complex preparation processes and the requirement for specialized equipment like xenon lamps or microwaves during catalysis [19,20,21,22].

ZnO is an excellent photocatalyst due to its high electron mobility and fast photogenerated electron transfer rate. However, its wide bandgap restricts light absorption to the UV region. Conversely, ZnFe_2_O_4_ has a narrower bandgap and strong visible light absorption capability, but suffers from rapid electron–hole recombination and low photocatalytic activity. In this study, a ZnO/ZnFe_2_O_4_ composite was synthesized via a simple and feasible hydrothermal method. Interface engineering was employed to enhance the visible-light photocatalytic activity, and the corresponding CO_2_ reduction performance was systematically evaluated [23,24,25,26].

## 2. Results and Discussion

### 2.1. Interface Structure Construction and Physicochemical Properties

Figure 1a presents the XRD patterns of the synthesized materials. For pure ZnFe_2_O_4_, the characteristic diffraction peaks observed at 2θ = 29.9°, 35.3°, 42.8°, 53.1°, 56.6°, and 62.2° can be assigned to the (220), (311), (400), (422), (511), and (440) crystal planes, respectively. These peaks match well with the standard pattern for cubic spinel ZnFe_2_O_4_ (JCPDS No. 79-1105), confirming its phase purity and structure. In the XRD patterns of the ZnO/ZnFe_2_O_4_ composites, additional distinct diffraction peaks appear at 2θ = 31.8°, 34.4°, 36.2°, and 67.9°. These peaks are indexed to the (100), (002), (101), and (112) planes of hexagonal wurtzite ZnO (JCPDS No. 36-1451), confirming the successful formation of the composite [14,15]. Alongside these ZnO peaks, the characteristic peaks of the cubic spinel ZnFe_2_O_4_ phase (e.g., (311), (400), (422), (511), (440)) remain present. More importantly, the composite exhibits notable structural improvements in the ZnFe_2_O_4_ phase compared to pure ZnFe_2_O_4_. Among these, the crystallinity of ZnO/ZnFe_2_O_4_ is significantly enhanced, as evidenced by the markedly reduced full width at half maximum (FWHM) of the (311) peak from 0.84 to 0.27, accompanied by a substantial increase in peak intensity. These changes suggest improved crystallite growth and structural ordering within the composite, as summarized in Appendix A. Applying Scherrer’s formula to this peak reveals a substantial increase in the average crystallite size of ZnFe_2_O_4_, from approximately 10.7 nm in the pure phase to 32.6 nm in the composite (Table 1). The diffraction peaks associated with ZnFe_2_O_4_ in the composite become noticeably sharper. These observed changes-reduced FWHM (increased crystallite size), lattice contraction, and peak sharpening-collectively indicate enhanced crystallinity and reduced lattice strain within the ZnFe_2_O_4_ phase in the composite. These results suggest that the presence of ZnO induces interfacial tensile strain and simultaneously promotes the growth of ZnFe_2_O_4_ crystallites [10,17,19]. The presence of ZnO promotes stress release at the interface, facilitates the formation of a well-defined heterojunction, lowers the surface energy, and drives the orderly growth of ZnFe_2_O_4_ crystallites. The successful integration of ZnO with ZnFe_2_O_4_, leading to these distinct structural modifications, is thus clearly evidenced by the XRD analysis.

Figure 1b displays the N_2_ adsorption-desorption isotherms along with the corresponding pore size distribution curves for both ZnFe_2_O_4_ and ZnO/ZnFe_2_O_4_ samples. The isotherm for pure ZnFe_2_O_4_ is classified as type IV, showing an H_1_-type hysteresis loop occurring at relative pressures (P/P_o_) between 0.9 and 1.0. This characteristic loop signifies the occurrence of capillary condensation, and its steep profile suggests the presence of relatively uniform cylindrical mesopores. The specific surface area (SSA) and total pore volume of ZnFe_2_O_4_ were determined to be 115.97 m^2^/g and 0.137 cm^3^/g, respectively. The pore size distribution (PSD) analysis Appendix A indicates that the pores are predominantly concentrated within the 1–10 nm range, with an average pore size of 4.73 nm. The ZnO/ZnFe_2_O_4_ composite also displays a type IV isotherm with an H_1_-type hysteresis loop (P/P_o_ = 0.9–1.0). Notably, the hysteresis loop appears steeper and narrower compared to that of pure ZnFe_2_O_4_, suggesting improved pore uniformity due to the homogeneous distribution of ZnFe_2_O_4_ nanoparticles on ZnO. However, a significant alteration is evident in the PSD curve of ZnO/ZnFe_2_O_4_. While mesopores (1–10 nm) remain present, a distinct new peak emerges in the 50–80 nm range, indicative of macropore formation. This shift is presumably attributable to the formation of interstitial voids resulting from the stacking of ZnFe_2_O_4_ nanoparticles on the lamellar ZnO structures with varied thicknesses during the multi-phase modification process. Consequently, the composite exhibits a significantly reduced SSA and an increased average pore size of 33.44 nm. The coexistence of mesopores and macropores, along with the larger average pore size in ZnO/ZnFe_2_O_4_, may facilitate the diffusion of reactant molecules within the catalyst structure [22].

As evidenced by the FT-IR spectrum (Figure 1c), the broad absorption band centered at 3385 cm^−1^ in ZnFe_2_O_4_ is associated with the O-H stretching vibration of adsorbed surface water. The distinct medium-intensity peak at 561 cm^−1^ is assigned to the Fe-O stretching vibration, confirming the successful synthesis of ZnFe_2_O_4_. In the spectra of the ZnO/ZnFe_2_O_4_ composites, the Fe-O stretching vibration remains clearly visible at 561 cm^−1^. Additionally, a distinct peak appears at 440 cm^−1^, assigned to the Zn-O stretching vibration within the tetrahedral sites of ZnO. The presence of both characteristic Zn-O and Fe-O vibration bands, coupled with the absence of other significant impurity peaks, confirms the successful synthesis of high-purity ZnO/ZnFe_2_O_4_ composites.

Raman spectra (Figure 1d) further reveal the coexistence of spinel-type ZnFe_2_O_4_ (A_1_g mode at 607 cm^−1^) and wurtzite ZnO (E_2_(high) mode at 441 cm^−1^) in the composite. Notably, both peaks exhibit significant sharpening and blue shifts (Δν = +36 cm^−1^ for A_1_g, Δν = +17 cm^−^^1^ for E_2_(high)), primarily resulting from interfacial lattice strain and strengthened Zn-O-Fe bonding at the heterojunction interface.

These observations are strongly supported by XRD analysis (Figure 1a), which shows lattice contraction, peak sharpening, and enhanced intensity, collectively indicating that interfacial engineering optimizes crystallinity while introducing controlled microstrain [23].

As shown in the TEM image of the pristine ZnFe_2_O_4_ (Figure 2a), the material consists of an aggregation of quasi-spherical particles. These particles possess a small grain size, predominantly within the 6–12 nm range. High-resolution imaging identifies distinct lattice fringes with a d-spacing of 0.254 nm, corresponding to the (311) plane of ZnFe_2_O_4_. Figure 2b displays the TEM image of the ZnO/ZnFe_2_O_4_ composite. In contrast to the pure phase, the composite exhibits a ZnO substrate decorated with well-dispersed ZnFe_2_O_4_ nanoparticles, with particle sizes ranging from approximately 30 to 150 nm. Well-defined lattice fringes are observed, with spacings of 0.260 nm and 0.254 nm, assignable to the (002) plane of ZnO and the (311) plane of ZnFe_2_O_4_, respectively. These clear lattice resolutions indicate high crystallinity within the composite, consistent with the sharpening of ZnFe_2_O_4_ diffraction peaks observed by XRD. The TEM results confirm the successful construction of a ZnO/ZnFe_2_O_4_ composite with intimate interfacial contact. The distinct lattice mismatch (d = 0.260 nm vs. d = 0.254 nm) and the observed interfacial coupling provide clear evidence for the formation of a well-coupled interface between the ZnO and ZnFe_2_O_4_ phases [27].

XPS measurements were conducted to examine the chemical states and interfacial characteristics of the samples. The survey scan (Figure 3a) verifies the existence of Zn, Fe, O, and C. In the Fe 2p region (Figure 3c), the peaks observed at 722.7 eV (Fe 2p_1/2_) and 711.0 eV (Fe 2p_3/2_) are indicative of Fe^2+^ and Fe^3+^ species, respectively, consistent with the Fe-O and Fe-O-Zn bonding environments in ZnFe_2_O_4_. Notably, in the ZnO/ZnFe_2_O_4_ composite, the Fe 2p peaks exhibit a positive shift of approximately 0.8 eV, while the Zn 2p peaks show a negative shift of about 1.4 eV. The Zn 2p spectrum of pristine ZnFe_2_O_4_ (Figure 3b) shows binding energies of 1021.32 eV (Zn 2p_3/2_) and 1044.38 eV (Zn 2p_1/2_), characteristic of Zn^2+^. In the ZnO/ZnFe_2_O_4_ composite, these peaks shift negatively to 1019.88 eV and 1042.94 eV, respectively. The large negative shift in Zn 2p (1.4 eV) exceeds typical heterojunction values (0.1−0.5 eV), suggesting contributions from interfacial dipole formation and local coordination environment changes in addition to charge transfer. These opposing shifts suggest a redistribution of electron density at the interface, consistent with electron transfer from ZnFe_2_O_4_ to ZnO. Quantitative analysis of XPS-derived elemental molar ratios reveals Zn and Fe ratios of ≈0.7:1 and 3.0:1, respectively Appendix A. The Fe ratio (3.0:1) closely corresponds to the CO/CH_4_ production rate ratio, providing a quantitative correlation between electronic structure modification and catalytic performance [19,22]. The observed increase in electron density around Zn atoms further supports this charge transfer mechanism. The O 1s spectrum (Figure 3d) exhibits three contributions: lattice oxygen in Zn-O (528.28 eV), Fe-O (530.18 eV), and surface-adsorbed water (532.08 eV). After the formation of the composite, the relative concentration of oxygen vacancies (associated with the 530.18 eV peak) increases from 21.3% to 25.3% (Table 2), providing more active sites for CO_2_ reduction. These results collectively demonstrate strong interfacial electronic interaction between ZnO and ZnFe_2_O_4_, which enhances the material’s catalytic properties [28].

### 2.2. Photoelectric Characteristics and Charge Separation Effect

Figure 4a–c presents the UV-visible diffuse reflectance spectra (DRS) of ZnFe_2_O_4_ and ZnO/ZnFe_2_O_4_ composites. The ZnO/ZnFe_2_O_4_ sample exhibits significantly enhanced light absorption across both UV and visible regions (500−700 nm) compared to pristine ZnFe_2_O_4_, along with a broader visible-light response range. This pronounced absorption enhancement indicates improved photocatalytic potential. The optical bandgaps (Eg) determined from the absorption onset (λg) via the tangent method (Figure 4b) are 1.96 eV for ZnFe_2_O_4_ and 2.12 eV for ZnO/ZnFe_2_O_4_, calculated using Eg = 1240/λg (eV). The observed Eg widening in the composite is attributed to interfacial lattice strain (consistent with the Raman blue-shift). While bandgap widening typically reduces photoabsorption, the introduction of ZnO (Eg ≈ 3.2 eV) extends the UV response. Importantly, the net enhancement in visible-light absorption originates from two synergistic effects: an increase in oxygen vacancies introducing defect states within the bandgap, as corroborated by XPS O 1s analysis, and interfacial engineering-induced band tailing that broadens the optical response range [25,27,28].

The Mott-Schottky curves (Figure 4c) exhibit positive slopes for both materials, confirming n-type semiconductor behavior [29]. The flat-band potential (E_FB_) approximates the conduction band potential (E_CB_) within ±0.1 eV for n-type semiconductors. The tangent extrapolation method yields E_CB_ values of −0.96 V (vs. NHE) for ZnFe_2_O_4_ and −0.75 V (vs. NHE) for ZnO Appendix A. Using the relation E_CB_ = E_VB_ − Eg [30], the valence band potentials (E_VB_) are calculated as +1.0 V and +2.44 V (vs. NHE) for ZnFe_2_O_4_ and ZnO, respectively.

Based on the band alignments, a type-II heterojunction is formed between ZnO and ZnFe_2_O_4_. This configuration promotes the transfer of photogenerated electrons from the conduction band of ZnFe_2_O_4_ (−0.96 eV) to that of ZnO (−0.75 eV), thereby facilitating CO_2_ reduction on the ZnO surface. Simultaneously, photogenerated holes migrate from the valence band of ZnO (+2.44 eV) to that of ZnFe_2_O_4_ (+1.0 eV), where oxidation reactions occur (Figure 4d). This charge separation mechanism is further supported by the enhanced photocurrent response (Figure 4e–g) and reduced charge transfer resistance observed in EIS measurements [31].

The efficient spatial separation of electrons and holes suppresses recombination, as confirmed by the 15.1% decrease in PL intensity (Figure 4f, Appendix A). The synergistic effect of improved charge separation and widened visible-light absorption collectively enhances the photocatalytic CO_2_ reduction performance of the ZnO/ZnFe_2_O_4_ composite.

### 2.3. Photocatalytic Performance

The photocatalytic CO_2_ reduction performance was evaluated in a solid-gas phase reaction system containing CO_2_ and water vapor under irradiation from a 300 W xenon lamp (λ ≥ 420 nm). For each test, 20 mg of catalyst was evenly spread on a sample holder. The reaction atmosphere, consisting of high-purity CO_2_ and water vapor, was introduced by injecting 0.3 mL of deionized water into the reactor as the proton source. The products were analyzed using an online gas chromatograph (GC9790II) equipped with a flame ionization detector (FID) and a thermal conductivity detector (TCD), using high-purity argon as the carrier gas.

As shown in Figure 5, the ZnFe_2_O_4_ sample produced no detectable CH_4_ or CO during the initial 4 h. Trace amounts of CH_4_ (0.016−0.024 μmol·g^−1^) and CO (0.018−0.035 μmol·g^−1^) were observed between 5 and 6 h.

In contrast, the ZnO-decorated ZnFe_2_O_4_ composite exhibited significantly enhanced photocatalytic activity. CH_4_ and CO were detected from the first hour onward, with cumulative yields reaching 0.103 μmol·g^−1^ and 0.208 μmol·g^−1^ after 6 h, respectively. These values correspond to 3.3-fold and 4.9-fold enhancements in CH_4_ and CO production compared to pure ZnFe_2_O_4_. Moreover, the composite demonstrated excellent stability over the reaction period.

The markedly improved activity and earlier onset of CO_2_ reduction indicate that interface engineering through ZnO decoration plays a critical role in facilitating catalytic performance. The modified interface provides abundant active sites, enhances CO_2_ adsorption and activation, and promotes efficient charge separation by suppressing the recombination of photoinduced charge carriers throughout the ZnFe_2_O_4_ matrix. These synergistic effects collectively contribute to the superior CO_2_ reduction performance of the ZnO/ZnFe_2_O_4_ composite [32].

### 2.4. Mechanism of Photocatalytic CO_2_ Reduction

As shown in Figure 6a, ZnFe_2_O_4_ exhibits characteristic -OH stretching vibrations in the range of 3750–3550 cm^−1^, along with signals at 2356 cm^−1^ and 2342 cm^−1^ corresponding to physically adsorbed CO_2_. The intensity of these adsorption peaks increases over time; however, no clear evidence of CO_2_ reduction intermediates or products is observed within 30 min of light irradiation, indicating limited photocatalytic activity. This result is consistent with the delayed production of CO and CH_4_ until the 5th hour during activity tests.

In contrast, the ZnO/ZnFe_2_O_4_ catalyst shows enhanced signals associated with -OH (3750–3550 cm^−1^) and CO_2_ (2356 cm^−1^ and 2342 cm^−1^) along with the rapid appearance of reduction intermediates under illumination, thereby facilitating the subsequent photocatalytic reduction process. As illustrated in Figure 6b, the appearance of a formate (HCOO^−^) peak at 1036 cm^−1^, as well as signals corresponding to key C_1_ pathway intermediates—such as *CHO at 1050, 1065, and 1078 cm^−1^, *CH_3_O at 1310 and 2971 cm^−1^, and formyl (*CHO) at 1700 cm^−1^—confirms active carbon hydrogenation. The emergence of a *CH_3_ peak at 2881 cm^−1^ and the gradual intensification of the 2971 cm^−1^ peak after 12 min indicate the formation of C-H-bonded species, suggesting CH_4_ generation (the adsorbed methyl vibration peak of CH_4_) [33]. Additionally, intermediates critical for CO formation are identified: *COOH at 1373 cm^−1^ and 1541–1714 cm^−1^, HCO_3_^−^ at 1391 and 1430 cm^−1^ (protonated precursor), and weakly adsorbed *CO at 2078 cm^−1^
Appendix A, which readily desorbs to facilitate CO release [34].

Time-dependent analysis of these characteristic peaks revealed formation rate constants of 0.081 mV·s/min for the CO pathway and 0.049 mV·s/min for the CH_4_ pathway (Figure 6c and Appendix A), indicating that the CO formation rate is 1.65 times faster than that of CH_4_. This kinetic result aligns well with the product distribution observed in photocatalytic tests (Section 2.3).

These results demonstrate that the ZnO/ZnFe_2_O_4_ interface promotes the formation and stabilization of reduction intermediates, substantially enhancing the CO_2_ photocatalytic reduction activity compared to ZnFe_2_O_4_ alone.

Based on the in situ FTIR evidence, two primary reaction pathways are proposed for product formation (Figure 6d): the CO pathway via *COOH intermediate and the CH_4_ pathway through progressive hydrogenation of *CH_3_O species.

CH_4_ formation may proceed via:

CO_2_ → HCOO^−^ → *CHO → *CH_3_O → CH_4_

CO_2_ → HCOO^−^ → *CHO → *CH_3_ → CH_4_

For CO production, the routes may proceed via:

CO_2_ → *CO_2_ → *CO → CO

CO_2_ → *CO_2_ → *COOH → HCO_3_^−^ → *CO → CO

In conclusion, the introduction of ZnO facilitates interfacial engineering that enhances CO_2_ adsorption, promotes key intermediate formation, and accelerates reaction kinetics, providing valuable insights for the rational design of efficient CO_2_ reduction photocatalysts [35].

## 3. Experimental Section

### 3.1. Materials

Detailed materials are provided in Appendix A.

### 3.2. Preparation of ZnFe_2_O_4_ and ZnO Microspheres via Solvothermal Method

ZnFe_2_O_4_ was synthesized via a solvothermal method. Briefly, Zn(NO_3_)_2_·6H_2_O and Fe(NO_3_)_3_·9H_2_O were dissolved in ethylene glycol with a molar ratio of Zn:Fe = 1:2. ZnO was prepared using a procedure similar to that of ZnFe_2_O_4_, but without the addition of Fe(NO_3_)_3_·9H_2_O. To adjust the alkalinity of the solution, 20 mL of NaOH aqueous solution (40 g/L) was added under continuous stirring. Detailed materials are provided in Appendix A.

### 3.3. Preparation of ZnO/ZnFe_2_O_4_ Composite via Sodium Hydroxide-Assisted Method

The ZnO/ZnFe_2_O_4_ composite was synthesized with a nominal Zn:Fe = 1.5:2. Similarly, 20 mL of NaOH solution (40 g/L) was introduced to maintain alkaline conditions during the synthesis. The resulting product was collected, washed, and dried for further characterization. Detailed materials are provided in Appendix A.

### 3.4. Catalysts Characterization

Detailed materials are provided in Appendix A.

### 3.5. Photocatalytic CO_2_ Reduction Reaction

Detailed materials of the photocatalytic CO_2_ reduction reaction were provided in Appendix A.

## 4. Conclusions

In this study, a one-step modification strategy was developed to decorate ZnFe_2_O_4_ with ZnO nanoparticles, forming a unique interfacial architecture that significantly enhances visible-light photocatalytic CO_2_ reduction. Comprehensive multiscale characterization reveals the following key insights:(1)Interfacial Optimization of Crystalline Structure: XRD and TEM analyses indicate that the presence of ZnO promotes the growth of ZnFe_2_O_4_ crystallites, increasing the average grain size from 10.7 nm to 32.6 nm. This is accompanied by a sharpening of the (311) diffraction peak, suggesting improved crystallinity [36]. Lattice parameter analysis reveals interfacial tensile strain (a reduction in the lattice constant *a* from 8.452 Å in pure ZnFe_2_O_4_), further supported by a Raman blue shift in the A_1_g mode (+36 cm^−1^).(2)The opposite XPS binding energy shifts of Zn 2p (−1.4 eV) and Fe 2p (+0.8 eV) suggest an interfacial electron transfer from ZnFe_2_O_4_ to ZnO, leading to an increased electron density around Zn and a decreased density around Fe. Mott–Schottky analysis confirms that the conduction band potential of ZnO is −0.75 eV while that of ZnFe_2_O_4_ is −0.96 eV, which is consistent with the formation of a Type-II heterojunction. Additionally, the oxygen vacancy concentration increases to 25.3% (from O 1s spectra), which facilitates extended visible-light absorption.(3)Enhanced Charge Separation and Transport: The built-in electric field formed at the Type-II heterojunction interface promotes directional migration of photogenerated electrons, resulting in a two-fold increase in photocurrent intensity (Figure 4e) and a 15.1% reduction in PL intensity (Figure 4f). Electrochemical impedance spectroscopy (EIS) shows reduced charge transfer resistance, collectively contributing to suppressed electron–hole recombination and prolonged carrier lifetime.(4)Synergistic Effects of Interface Engineering: The ZnO/ZnFe_2_O_4_ system overcomes the inherent limitations of the individual components (e.g., rapid charge recombination in ZnFe_2_O_4_ and narrow light response of ZnO). Through the Type-II band alignment, the composite achieves synergistic broad visible-light absorption (Eg = 2.12 eV) and strong reduction capacity (E_CB_ = −0.96 eV), leading to a dramatic improvement in CO_2_ photoreduction performance. After 6 h of visible-light irradiation, the production rates of CH_4_ and CO increased by 3.3 and 4.9 times, respectively, compared to pure ZnFe_2_O_4_.

## Figures and Tables

**Figure 1 molecules-30-03980-f001:**
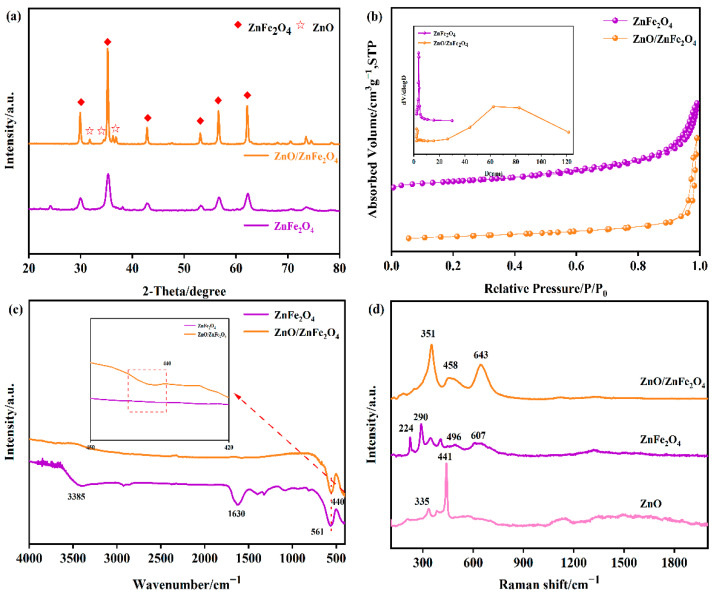
(**a**) XRD patterns of the synthesized ZnFe_2_O_4_, ZnO/ZnFe_2_O_4_, (**b**) N_2_ adsorption-desorption isotherms and PSD curves (inset), (**c**) FT-IR spectra of samples ZnFe_2_O_4_ and ZnO/ZnFe_2_O_4_ catalysts, (**d**) Raman spectra of samples ZnFe_2_O_4_ and ZnO/ZnFe_2_O_4_ catalysts with different nanomorphologies.

**Figure 2 molecules-30-03980-f002:**
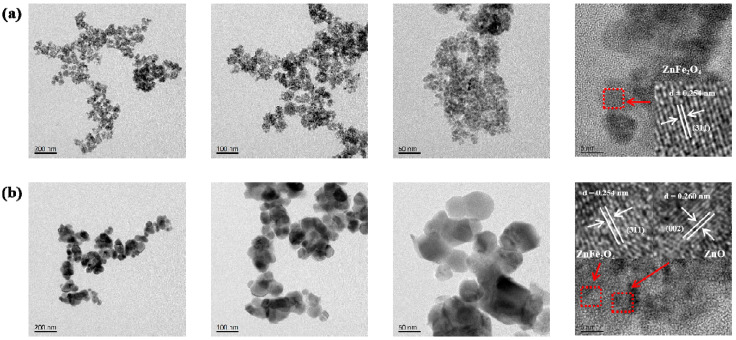
TEM and HRTEM images of (**a**) ZnFe_2_O_4_, (**b**) ZnO/ZnFe_2_O_4_.

**Figure 3 molecules-30-03980-f003:**
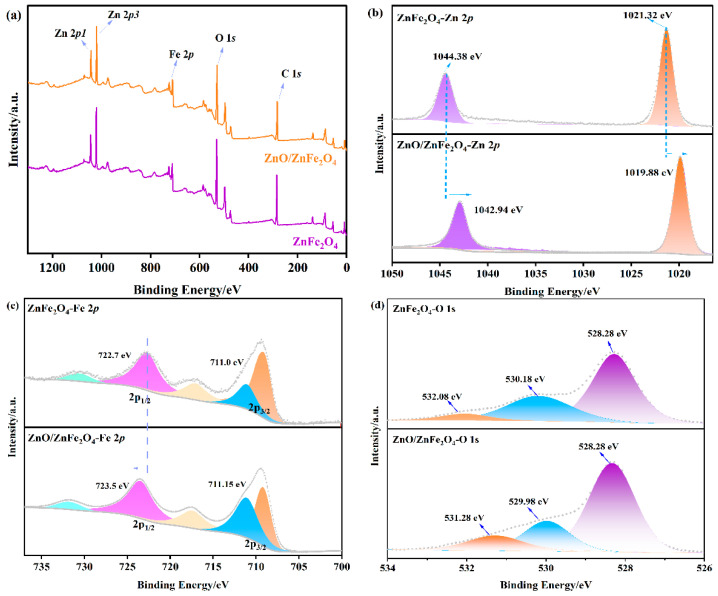
XPS of (**a**) survey spectra, (**b**) Fe 2p, (**c**) Zn 2p, and (**d**) O 1s of ZnFe_2_O_4_ and ZnO/ZnFe_2_O_4_.

**Figure 4 molecules-30-03980-f004:**
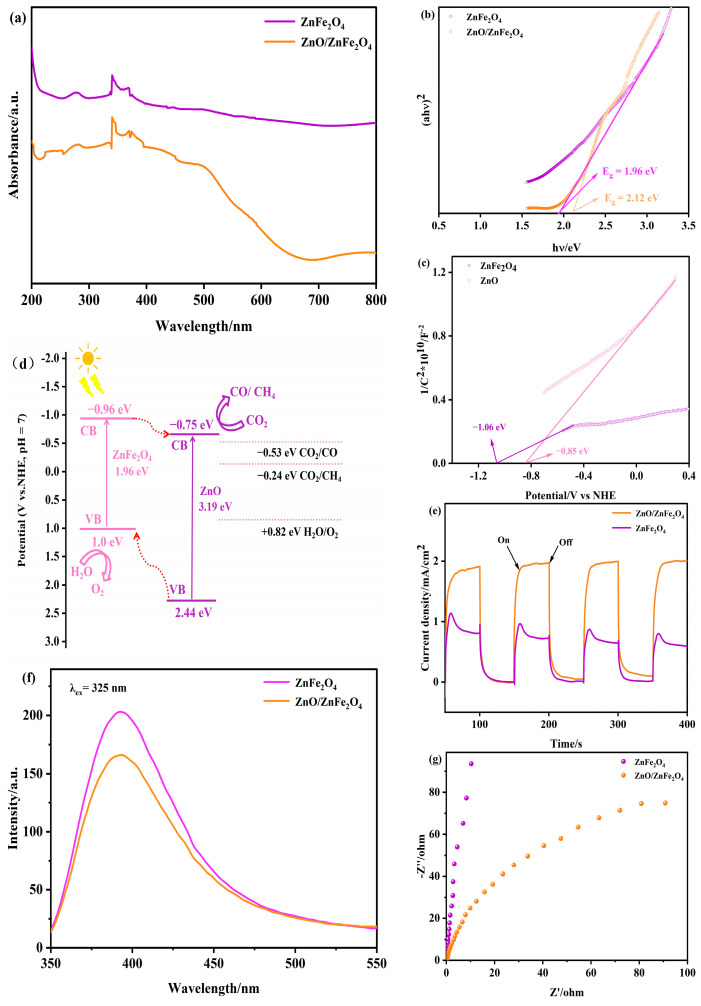
(**a**–**c**) UV-vis DRS, (**d**) proposed photocatalytic mechanism, (**e**) transient photocurrent response plots, (**f**) PL, and (**g**) EIS of ZnO and ZnO/ZnFe_2_O_4_.

**Figure 5 molecules-30-03980-f005:**
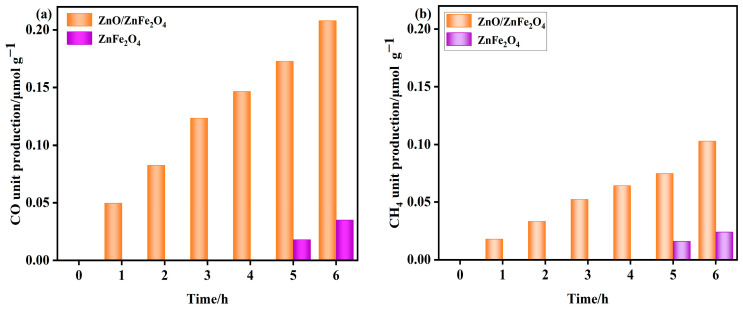
(**a**,**b**) Product unit yields of ZnO and ZnO/ZnFe_2_O_4_ catalysts (unit yield of CO and CH_4_).

**Figure 6 molecules-30-03980-f006:**
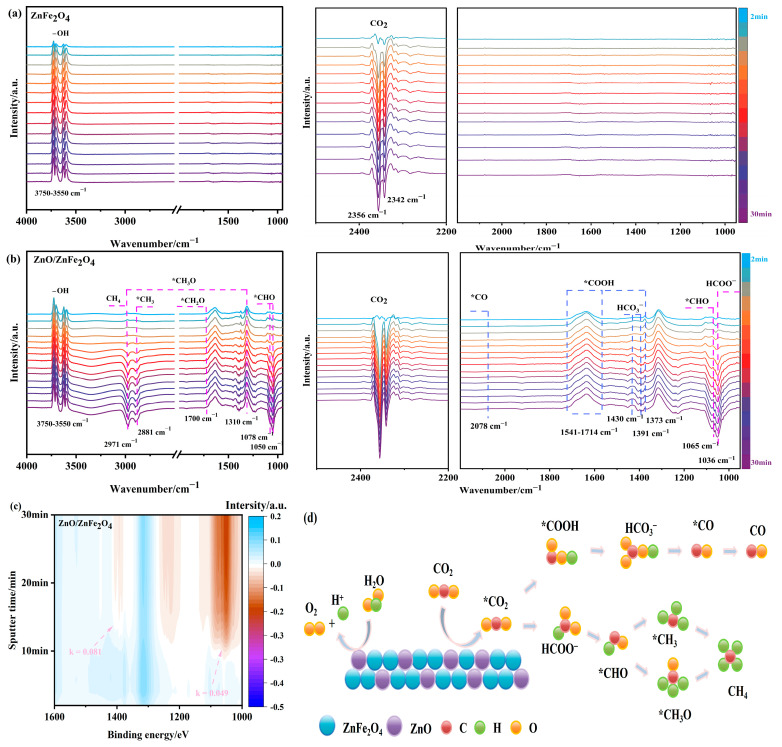
(**a**–**c**) In situ FTIR and (**d**) the possible processes for CO_2_ reduction of ZnO/ZnFe_2_O_4_.

**Table 1 molecules-30-03980-t001:** XRD structural parameter of samples ZnO and ZnFe_2_O_4_.

Samples	Crystal Size (nm)	Crystal Parameter (Å)			Standard Card
		a	b	c	
ZnO	1.2	3.250	3.250	5.207	JCPDS No. 36-1451
ZnFe_2_O_4_	10.7	8.466	8.466	8.466	JCPDS No. 79-1105
ZnO/ZnFe_2_O_4_	32.6	3.250	3.250	5.207	JCPDS No. 89-7102

**Table 2 molecules-30-03980-t002:** The chemical state of Zn, Fe and O in the catalysts.

Samples	O Atomic/%			Atomic/%		
	O_latt_	O_ads_	O_w_	Zn	Fe	O
ZnFe_2_O_4_	66.2	21.3	12.5	23.8	40.2	36.0
ZnO/ZnFe_2_O_4_	68.6	25.3	6.1	36.4	16.4	47.2

## Data Availability

Data are contained within the article.

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
