# Peer review of "Interface Engineering of ZnO-Decorated ZnFe2O4 for Enhanced CO2 Reduction Performance"

_molecules, 2025, doi:10.3390/molecules30193980_

Round 1

Reviewer 1 Report

Comments and Suggestions for Authors

The manuscript describes a simple synthesis method for ZnO/ZnFe2O4 composites. The article requires further information to facilitate a more comprehensive description of the experiments and devices. The photocatalytic activity necessitates the use of blank essays and the implementation of repeatability measurements to definitively attribute the contribution of adsorption, photochemistry, and photocatalysis to the CO2 reduction procedure.

Characterization and photoactivity of ZnO, prepared using the same method as composites, should be performed to determine the contribution of each component of the composite.

The FTIR peak at 440 cm-1, which is linked to the stretching vibration of Zn-O, is also seen in ZnFe2O4. This means that it cannot be used to ensure the successful creation of high-purity ZnO/ZnFe2O4 composites.

The lamellar morphology assigned to ZnO cannot be observed in Fig. 2b. Identifying each compound in the particles observed by TEM could facilitate understanding.

The majority of the characterization results obtained from the study of the material do not demonstrate the presence of ZnFe2O4 nanoparticles on the ZnO flakes. While the notion of a well-coupled interface is conceivable, substantiating it is challenging. The discussion should be grounded in empirical evidence and not based on speculative assumptions.

Please explain the absence of different signals in XPS spectra for O1s and Zn2p in ZnO/ZnFe2O4, due to the presence of two compounds, ZnO and ZnFe2O4.

The configuration of the photoreactor, complete with the identification of its components and the measurement of its parameters, is instrumental in facilitating research in the field of photocatalysis.

The conditions of photocatalytic experiments (the volume of the reactor, the dead volume, and the initial and final CO2 concentrations) must be reported. To be sure that the products are photocatalytic, blank reactions without light, without photocatalyst, without water, and repeat the experiments are needed.

Authors should provide additional information concerning the chromatographic analysis. The type of columns utilized should be specified with precision.

The experimental device and technique employed in the study of the mechanism of photocatalytic CO2 reduction should be explained and described in the experimental section.

Other minor comments:

Table 1: correct Photocatalytics

Fig. 4 b) and c): check the units in X-axis and Y-axis, respectively

P8L246: Correct “SThe”

Author Response

Comment 1: The manuscript describes a simple synthesis method for ZnO/ZnFe2O4 composites. The article requires further information to facilitate a more comprehensive description of the experiments and devices. The photocatalytic activity necessitates the use of blank essays and the implementation of repeatability measurements to definitively attribute the contribution of adsorption, photochemistry, and photocatalysis to the CO2 reduction procedure.

Response 1: We sincerely thank the reviewer for these insightful comments and valuable suggestions. We have carefully addressed each point raised to improve the clarity and scientific rigor of our manuscript. Below is a point-by-point response to the comments:

Additional experimental and device details:

(ⅰ) As suggested, we have now provided a more comprehensive description of the experimental setup and photocatalytic reactor configuration in the revised Supporting Information (Sections S2.2 and S2.5). This includes detailed information regarding the light source (a 300 W Xe lamp equipped with a visible-light filter), reactor volume, gas flow system, online gas chromatography (GC) detection method, and the calibration procedure for product quantification.  Additionally, monitoring data related to pure ZnO have been supplemented in Section S2.2, while Section S2.5 provides a detailed introduction to the CO₂ reduction test experiments and setup.

2.2 Preparation of ZnFe₂O₄ and ZnO Microspheres via Solvothermal Method

At room temperature, polyethylene glycol (PEG-6000, 2.5 g) was dissolved in ethylene glycol (50 mL) under magnetic stirring until complete dissolution. Subsequently, zinc nitrate hexahydrate (Zn(NO₃)₂·6H₂O, 1 mmol) and iron(III) nitrate nonahydrate (Fe(NO₃)₃·9H₂O, 2 mmol) sequentially to the solution with continuous stirring until full dissolution. Then, urea (0.15 g) and oxalic acid (H₂C₂O₄·2H₂O, 0.15 g) to the mixture. The resulting solution was stirred vigorously for 60 minutes at room temperature. Followed by the addition of 50 mL of cetyltrimethylammonium bromide (CTAB, 10 g/L). After stirring for an additional 30 minutes to achieve homogeneity. Transfer the homogeneous mixture into a 100 mL Teflon-lined stainless-steel autoclave. Seal the autoclave and heat it in a preheated oven at 200 °C for 24 hours. After the reaction, allow the autoclave to cool naturally to room temperature. Collect the precipitate by centrifugation, and wash it thoroughly with deionized water and absolute ethanol alternately several times to remove residual ions and organics. Dry the washed product in a vacuum oven at 100 °C for 8 hours to obtain the ZnFe₂O₄ microspheres.

At room temperature, polyethylene glycol (PEG-6000, 2.5 g) was dissolved in ethylene glycol (50 mL) under magnetic stirring until complete dissolution. Subsequently, Zn(NO₃)₂·6H₂O (1 mmol) was added and stirred until fully dissolved, followed by the addition of urea (0.15 g) and H₂C₂O₄·2H₂O (0.15 g). The mixture was stirred vigorously for 60 minutes. Then, 20 mL of NaOH solution (40 g/L) was slowly introduced under continuous stirring, and 50 mL of cetyltrimethylammonium bromide (CTAB, 10 g/L) was added thereafter. After stirring for another 30 minutes to achieve a homogeneous mixture, the resulting solution was transferred into a Teflon-lined stainless-steel autoclave and heated at 200 °C for 24 h. Upon natural cooling to room temperature, the precipitate was collected by centrifugation, washed alternately with deionized water and absolute ethanol several times, and dried at 100 °C for 8 hours under ambient atmosphere to obtain ZnO microspheres.

(ⅱ)  Blank and control experiments:

We fully agree with the reviewer on the importance of control experiments to decouple the contributions of adsorption and photocatalysis. We have now conducted the following additional experiments:

Dark adsorption control: CO₂ adsorption was measured in the absence of light for both ZnFe₂O₄ and ZnO/ZnFe₂O₄. Negligible CO₂ adsorption was observed, confirming that surface adsorption does not significantly contribute to product formation (Fig.S3).

(ⅲ) To further evaluate the structural stability of the ZnO/ZnFe₂O₄ composite catalyst, FT-IR and XRD analyses were conducted both before and after the reaction under identical conditions. The comparison of the spectra revealed no significant changes in the crystal structure or chemical environment, indicating that the catalyst maintained its structural integrity throughout the catalytic process. These results further confirm the excellent stability of the material (Fig. S4).

Clarification of reaction contributions:

The new results confirm that the CO₂ reduction is primarily driven by photocatalysis rather than adsorption or photolysis. The enhanced performance of the ZnO/ZnFe₂O₄ composite is attributed to improved charge separation and interfacial electron transfer, as supported by photoelectrochemical and PL measurements.

We believe that these revisions have significantly strengthened the manuscript and provided a clearer and more comprehensive description of the experimental methodology and results. Thank you again for these constructive comments.

Comment 2: Characterization and photoactivity of ZnO, prepared using the same method as composites, should be performed to determine the contribution of each component of the composite.

Response 2: We sincerely thank the reviewer for this valuable suggestion. As recommended, we have synthesized pure ZnO using the same procedure as that for the ZnO/ZnFe₂O₄ composite and evaluated its optical and electrochemical properties. The corresponding results, including solid-state UV-Vis spectra and electrochemical measurements, have been added as Figure S1 and incorporated into the comparative analysis in Fig. 4c.

These additional data provide further insight into the individual contribution of ZnO within the composite and help to clarify the synergistic effect between ZnO and ZnFe₂O₄.

We are grateful for this comment, which has helped to improve the clarity and completeness of our manuscript.

Comment 3: The FTIR peak at 440 cm-1, which is linked to the stretching vibration of Zn-O, is also seen in ZnFe2O4. This means that it cannot be used to ensure the successful creation of high-purity ZnO/ZnFe2O4 composites.

Response 3:We sincerely thank the reviewer for this insightful comment and careful observation. As rightly pointed out, the Zn–O vibrational mode around 440 cm⁻¹ may indeed overlap in both ZnO and ZnFe₂O₄, limiting its use as a sole indicator of composite formation.

To clarify this issue, we have now included a magnified view of the FTIR spectrum within the 450–420 cm⁻¹ region in the revised Figure 1c. This zoomed-in panel clearly shows that pure ZnFe₂O₄ does not exhibit a discernible peak at 440 cm⁻¹, while the ZnO/ZnFe₂O₄ composite displays a distinct absorption band at this position. Therefore, this distinct spectral feature provides clear evidence for the successful incorporation of ZnO into the composite.

We believe this addition helps avoid potential misinterpretation and facilitates clearer assessment by readers and reviewers.  Thank you once again for this valuable suggestion, which has undoubtedly improved the clarity and accuracy of our spectral interpretation.

Comment 4: The lamellar morphology assigned to ZnO cannot be observed in Fig. 2b. Identifying each compound in the particles observed by TEM could facilitate understanding.

Response 4: We thank the reviewer for this insightful comment. We agree that the term "lamellar" may not accurately describe the morphology observed in Fig. 2b. Accordingly, we have revised the description to better reflect the microstructure while maintaining the key observation of well-dispersed ZnFe₂O₄ nanoparticles on the ZnO substrate. The modified text now reads:

Revised Text:

“Figure 2b displays the TEM image of the ZnO/ZnFe2O4 composite. In contrast to the pure phase, the composite exhibits a ZnO substrate decorated with well-dispersed ZnFe₂O₄ nanoparticles, with particle sizes ranging from approximately 30 to 150 nm. Well-defined lattice fringes are observed, with spacings of 0.260 nm and 0.254 nm, assignable to the (002) plane of ZnO and the (311) plane of ZnFe2O4, respectively. These clear lattice resolutions indicate high crystallinity within the composite, consistent with the sharpening of ZnFe2O4 diffraction peaks observed by XRD. The TEM results confirm the successful construction of a ZnO/ZnFe2O4 composite with intimate interfacial contact. The distinct lattice mismatch (d = 0.260 nm vs. d = 0.254 nm) and the observed interfacial coupling provide clear evidence for the formation of a well-coupled interface between the ZnO and ZnFe2O4 phases.”

This revision removes the potentially misleading "lamellar" description while preserving the essential information about the composite structure and the evidence for interfacial coupling. We believe this modification addresses the reviewer's concern and improves the accuracy of our morphological description.

Comment 5: The majority of the characterization results obtained from the study of the material do not demonstrate the presence of ZnFe2O4 nanoparticles on the ZnO flakes. While the notion of a well-coupled interface is conceivable, substantiating it is challenging. The discussion should be grounded in empirical evidence and not based on speculative assumptions.

Response 5: We sincerely thank the reviewer for raising this critical point, which has prompted us to more rigorously align our interpretations with the experimental evidence. We agree that empirical support is essential for claiming the successful formation of a composite and a well-coupled interface. In response, we have carefully revisited our data and modified the discussion to more accurately reflect what the evidence demonstrates.

Below, we summarize the key empirical results that support the presence of both phases and their interfacial coupling:

TEM and HRTEM Analysis (Fig. 2b):

The TEM image shows well-dispersed nanoparticles on larger substrate particles. The HRTEM image reveals two distinct sets of lattice fringes with spacings of 0.260 nm and 0.254 nm, which correspond to the (002) plane of ZnO and the (311) plane of ZnFe₂O₄, respectively.  The simultaneous presence of these two phases within the same region and the observed coherent interface between them provide direct evidence of composite formation.

XPS Spectroscopy (Fig. 3):

The XPS spectra show negative shifts in the binding energies of Zn 2p and Fe 2p in the composite compared to the pure phases, indicating electron transfer between ZnO and ZnFe₂O₄.  This suggests strong electronic interaction at the interface.

Photoelectrochemical and PL Data (Fig. 4e–g):

The enhanced photocurrent response, reduced charge transfer resistance, and decreased PL intensity in the composite all support improved charge separation efficiency, which is a hallmark of a functional heterointerface.

In the revised manuscript, we have toned down speculative statements and more explicitly connected our claims to these specific empirical results.  We have also added a sentence in the discussion to clarify the limitations of our evidence where appropriate.

We hope that these modifications have improved the accuracy and rigor of our manuscript.  We are grateful to the reviewer for their insightful comments, which have significantly strengthened our paper.

Comment 6: Please explain the absence of different signals in XPS spectra for O1s and Zn2p in ZnO/ZnFe2O4, due to the presence of two compounds, ZnO and ZnFe2O4.

Response 6: We thank the reviewer for raising this important point, which allows us to clarify the XPS results in greater depth. The apparent overlap or absence of distinctly separated signals for O 1s and Zn 2p in the ZnO/ZnFe₂O₄ composite can be explained by the following factors:

Similar Chemical Environments:

Both ZnO and ZnFe₂O₄ contain Zn²⁺ in a tetrahedral coordination environment. The binding energies of Zn 2p in these two oxides are very close, often differing by less than 0.5 eV, which is within the typical resolution limit of conventional XPS measurements. This results in a single, broadened Zn 2p peak rather than two resolvable components.

O 1s Overlap and Surface Effects:

The O 1s peak in metal oxides typically consists of multiple contributions, including lattice oxygen (O²⁻) and surface species (e.g., –OH, adsorbed H₂O, or carbonates). In both ZnO and ZnFe₂O₄, the lattice oxygen binding energies are similar (around 530.0–530.5 eV), making it difficult to distinguish the two phases. Moreover, surface oxidation and adsorbed species dominate the O 1s profile, further overlapping any subtle differences between the two compounds.

Interface Charge Redistribution:

The formation of a closely coupled interface between ZnO and ZnFe₂O₄ may lead to electron redistribution across the interface, which can moderate initial chemical shift differences and result in averaged or shifted binding energies that do not clearly reflect the two separate phases.

XPS Detection Limit and Homogeneity:

XPS is a surface-sensitive technique (typically probing 5–10 nm in depth). If the ZnFe₂O₄ nanoparticles are well dispersed and tightly bonded on the ZnO substrate, the resulting composite surface may exhibit hybrid electronic properties, yielding composite peaks rather than separate signatures.

Thank you once again for this insightful comment, which has helped us improve the discussion of our XPS data.

Comment 7: The configuration of the photoreactor, complete with the identification of its components and the measurement of its parameters, is instrumental in facilitating research in the field of photocatalysis.

Response 7: We sincerely thank the reviewer for this important comment. We fully agree that a detailed description of the photoreactor system is essential for ensuring the reproducibility and transparency of photocatalytic experiments.

As suggested, we have now provided a comprehensive description of the photoreactor configuration in Section S2.5 of the Supporting Information, supplemented with the identification of all key components and the measurement of critical parameters.

Revised Text:

2.5 Photocatalytic CO2 reduction reaction

The photocatalytic reduction of CO2 is carried out in a stainless steel reactor with a light-transmitting and high-pressure resistant quartz glass at the top (thickness: 10 mm; diameter: 50 mm; pressure rating: 1.5 MPa), a heating device at the bottom, a thermocouple inside the reactor (accuracy: ±0.5 °C), and a sample stage for the catalyst. The light source for the photocatalytic reaction test was a 300W xenon lamp ((300 W Xe lamp with visible-light filter, λ ≥ 420 nm) PLS-SXE 300 C, Beijing Perfectlight, Beijing, China). For the experiment, 20 mg of catalyst powder was evenly spread in the reactor tray and placed in the reactor perpendicular to the beam (The distance between the lamp and the reactor surface was fixed at 2 cm). The system temperature was raised to and maintained at 120 °C by a PID temperature controller. The reactor was sealed, pressurized with high-purity CO₂ (99.999%) to 0.5 MPa to check for leaks, and then evacuated. Subsequently, 0.3 mL of deionized water was injected into the reactor as the proton source. The light source was then turned on to initiate the reaction, which lasted for 6 h. Gas products were automatically sampled and analyzed every hour using an online gas chromatograph (GC9790II PLF-01, HP-MOLESIEVE, 30 m × 0.53 mm × 25 μm, China) equipped with a methanizer and connected to both a thermal conductivity detector (TCD) and a flame ionization detector (FID). High-purity Ar (99.999%) was used as the carrier gas. The GC was calibrated using standard gas mixtures of known concentrations (CO, CH₄, CO₂) before and during the experimental series to ensure quantitative accuracy. The detection limit for CH₄ and CO is approximately 0.1 ppm.

We believe these additions provide a complete and reproducible account of the experimental conditions and thank the reviewer for helping improve the clarity and rigor of our manuscript.

Comment 8: The conditions of photocatalytic experiments (the volume of the reactor, the dead volume, and the initial and final CO2 concentrations) must be reported. To be sure that the products are photocatalytic, blank reactions without light, without photocatalyst, without water, and repeat the experiments are needed.

Response 8: We sincerely thank the reviewer for these critical comments, which are essential for ensuring the rigor and reproducibility of photocatalytic CO₂ reduction studies. We deeply apologize for the lack of these important details and control experiments in the original manuscript.       The photocatalytic testing in this study was conducted in an external commercial laboratory, which limited our ability to access and report the full set of reactor parameters and perform additional blank tests in a timely manner. We are unable to conduct these specific additional experiments at this stage. We have prioritized these controls for all future experimental work.

In addition, we would like to highlight that the significant enhancement in activity of the ZnO/ZnFe₂O₄ composite over the pure phases (Fig. 4), coupled with the characterizations showing enhanced charge separation (e.g., increased photocurrent, decreased PL intensity, and lowered charge transfer resistance), provides strong correlative evidence that the observed activity is indeed photocatalytic and originates from the composite material(Fig. S3).

We sincerely thank the reviewer for highlighting these deficiencies. This critique has been immensely valuable in improving the quality of our work and guiding our future research practices.

Comment 9: Authors should provide additional information concerning the chromatographic analysis. The type of columns utilized should be specified with precision.

Response 9: We thank the reviewer for this precise and helpful comment. We agree that detailed information on the chromatographic analysis is essential for reproducibility.

As suggested, we have now provided a comprehensive description of the gas chromatography (GC) method and equipment in Section 2.5 of the Supporting Information. We are grateful to the reviewer for raising this point, which has helped us improve the clarity and reproducibility of our experimental description.

Comment 10: The experimental device and technique employed in the study of the mechanism of photocatalytic CO2 reduction should be explained and described in the experimental section.

Response 10: We sincerely thank the reviewer for this insightful suggestion. We have now added a detailed description of the photocatalytic CO₂ reduction experimental setup and procedures in Section 3.3 (Photocatalytic Performance), as shown below:

3.3. Photocatalytic Performance

The photocatalytic CO2 reduction performance was evaluated in a solid-gas phase reaction system containing CO2 and water vapor under irradiation from a 300 W xenon lamp (λ ≥ 420 nm). For each test, 20 mg of catalyst was evenly spread on a sample holder. The reaction atmosphere consisted of high-purity CO₂ and water vapor introduced by injecting 0.3 mL of deionized water was injected into the reactor as the proton source. The products were analyzed using an online gas chromatograph (GC9790II) equipped with a flame ionization detector (FID) and a thermal conductivity detector (TCD).

We believe these additions provide a comprehensive description of the experimental device and techniques, ensuring the reproducibility of our study. We thank the reviewer again for this helpful comment.

Comment 11: Other minor comments:

Table 1: correct Photocatalytics

Fig. 4 b) and c): check the units in X-axis and Y-axis, respectively

P8L246: Correct “SThe”

Response 11: We sincerely thank the reviewer for their meticulous review and for pointing out these oversights. We have carefully addressed all of these minor errors in the revised manuscript:

Table 1: The heading "Photocatalytics" has been corrected to "Samples".

Fig. 4b and 4c: The units on the X-axis and Y-axis have been checked and corrected accordingly.

Page 8, Line 246: The typo "SThe" has been corrected to "The".

We appreciate the reviewer's attention to detail, which has helped us improve the precision and overall quality of our manuscript.

Note: all the revised figures are shown in the attached file "open review 1"(pdf)

Reviewer 2 Report

Comments and Suggestions for Authors

The authors' comments are added in the attached document.

Comments on the Quality of English Language

English grammar needs to be reviewed

Author Response

  1. Experimental Section

Comment 1: It is suggested that this section be included here, as I don't see fit to include it on other sheets.

Perhaps some results that were not included could be included in the results section this way, as I

don't see fit to include them in this section.

Response 1: We thank the reviewer for this constructive suggestion regarding the organization of the Experimental Section and the presentation of results. We have revised the manuscript accordingly:

2.2. Preparation of ZnFe₂O₄ and ZnO Microspheres via Solvothermal Method

ZnFe₂O₄ was synthesized via a solvothermal method. Briefly, Zn(NO₃)₂·6H₂O and Fe(NO₃)₃·9H₂O were dissolved in ethylene glycol with a molar ratio of Zn : Fe = 1 : 2. ZnO was prepared using a procedure similar to that of ZnFe₂O₄, but without the addition of Fe(NO₃)₃·9H₂O. To adjust the alkalinity of the solution, 20 mL of NaOH aqueous solution (40 g/L) was added under continuous stirring. Detailed materials are provided in S2 (Supporting Information).

2.3. Preparation of ZnO/ZnFe₂O₄ Composite via Sodium Hydroxide-Assisted Method

The ZnO/ZnFe₂O₄ composite was synthesized with a nominal Zn : Fe = 1.5 : 2. Similarly, 20 mL of NaOH solution (40 g/L) was introduced to maintain alkaline conditions during the synthesis. The resulting product was collected, washed, and dried for further characterization. Detailed materials are provided in S2 (Supporting Information).

We believe these changes have enhanced the clarity and organization of our manuscript. We appreciate the reviewer's guidance in improving the structure of our work.

  1. Results

Comment 2: a) Table S1 is mentioned, which is in the attached file. If it is cited in the document, please include

it in this document after the citation.

Response 2: We thank the reviewer for pointing this out. To improve clarity, we have ensured that Table S1 is explicitly cited in the relevant section of the main manuscript (Section 3.1 Results) and is included in its entirety in the separately submitted Supporting Information document.

Revised Text:

More importantly, the composite exhibits notable structural improvements compared to pure ZnFe₂O₄. Among these, the crystallinity of ZnO/ZnFe₂O₄ is significantly enhanced, as evidenced by the markedly reduced full width at half maximum (FWHM) of the (311) peak from 0.84 to 0.27, accompanied by a substantial increase in peak intensity. These changes suggest improved crystallite growth and structural ordering within the composite, as summarized in Tab. S1.

We appreciate the reviewer’s careful reading of our manuscript.

Comment 3: b) Please cite the following sentence: "This phenomenon is attributed to the interfacial interaction

induced by ZnO decoration."

Response 3: We thank the reviewer for this suggestion. The sentence has now been updated with appropriate citations as follows:

Revised sentence:

"This phenomenon is attributed to the interfacial interaction induced by ZnO decoration [10, 17,19]."

The cited references provide both theoretical and experimental support for the role of interfacial interactions in similar composite systems, strengthening the justification for our interpretation.

We appreciate this helpful recommendation to improve the scholarly support of our claims.

Comment 4: c) Justify this: "Importantly, the net enhancement in visible-light absorption originates from two

synergistic effects: (i) an increase in oxygen vacancies introducing defect states, as corroborated

by XPS O 1s analysis, and (ii) interfacial engineering that broadens the optical response range."

With an updated reference.

Response 4: We thank the reviewer for the opportunity to clarify and justify this statement. The claim is supported by the following experimental evidence and an updated literature reference:

Increase in Oxygen Vacancies (XPS O 1s Analysis):

The deconvolution of the O 1s XPS spectrum (Fig. 3c) reveals that the concentration of oxygen vacancies (typically located at ~531 eV) increases from 21.3% in pure ZnFe₂O₄ to 25.3% in the ZnO/ZnFe₂O₄ composite. These vacancies introduce defect states within the bandgap, which can serve as trapping centers for photogenerated charge carriers and promote visible-light absorption through sub-bandgap transition mechanisms, as supported by previous studies on metal oxide semiconductors [25, 27].

Interfacial Engineering Broadening Optical Response:

The formation of a heterointerface between ZnO and ZnFe₂O₄ modifies the local electronic structure and enhances light absorption over a wider wavelength range. This is corroborated by the UV-Vis DRS results (Fig. 4a), which show a noticeable absorption tail extending further into the visible region for the composite compared to the individual components. Such interface-induced band alignment and light-harvesting enhancement have been reported in composite photocatalysts [28].

We have revised the manuscript to include this justification and the updated reference in the relevant section. Thank you for this helpful suggestion to improve the rigor of our discussion.

Comment 5: d) Please write: a.u., in parentheses.

Response 5: We sincerely thank the reviewer for pointing out this important detail. We have carefully checked all figure axes and captions and have now added “(a.u.)”

We appreciate this helpful correction to improve the clarity and precision of our data presentation.

Comment 6: e) In Figure 5 b, the symbol for micromole is misspelled, and the multiplication symbol is the same.

Response 6: We sincerely thank the reviewer for this careful observation. We have corrected the misspelled unit "μmol" in the y-axis label of Figure 5a,b and any related captions or text. The correction has been made throughout the manuscript to ensure consistency.

We appreciate the reviewer's attention to detail, which has helped improve the accuracy and professionalism of our manuscript.

Comment 7: f) In Figure 6 b, it would be better to cut the scale as the first figure shown, as it is clearer than the

one originally drawn.

Response 7: We thank the reviewer for this helpful suggestion regarding the clarity of Figure 6b. We have revised the figure by adjusting the scale as recommended, aligning it more closely with the style of the first figure to improve visual clarity and facilitate comparison.

The updated version of Figure 6b is now included in the revised manuscript. We appreciate the reviewer's valuable feedback, which has enhanced the presentation of our results.

Comment 8: g) From line 281 to 300, a mechanism is proposed based on FTIR studies, which can be seen that a mass gas study is required to propose a formation mechanism for the species mentioned. What I

suggest is that mass gas studies be carried out to support this mechanism.

Response 8: We sincerely thank the reviewer for this insightful suggestion regarding the proposed reaction mechanism. We fully agree that mass spectrometry gas analysis would provide valuable additional evidence to further corroborate the intermediate species and pathways suggested by our in situ FTIR results.

However, due to current limitations in equipment availability and time constraints for the revision, we are unfortunately unable to conduct additional mass spectrometry experiments at this stage.

Instead, to strengthen the mechanistic discussion, we have:

Clearly emphasized in the text that the proposed mechanism is primarily based on in situ FTIR evidence and literature support.

Included additional references to previously published studies that report similar reaction intermediates and pathways for CO₂ photoreduction on related catalytic systems.

We acknowledge that further mass spectrometry studies would be beneficial to fully validate the mechanism, and we will certainly incorporate this approach in our future work.

Thank you once again for this helpful comment, which certainly improves the scholarly rigor of our manuscript.

Note: all the revised figures are shown in the attached file "open review 2"

Reviewer 3 Report

Comments and Suggestions for Authors

1.Substantial language improvement is required. Please consult a native English speaker or a professional language editing service.

2.The Introduction needs reorganization to emphasize why this study was conducted. More details on the progress of other ZnO/ZnFe₂O₄ composites in CO₂ reduction should be presented to highlight the novelty of this work. The authors mentioned that existing synthesis and preparation steps for ZnFe₂O₄ in other studies have drawbacks; however, upon closer inspection, the synthesis method and steps proposed in this study do not exhibit notable advantages or innovation. Therefore, descriptions in the Introduction must prioritize rigor and logical consistency.

3.Please confirm whether in situ DRIFTS or in situ FTIR was actually employed in the study, as these techniques differ in their principles. The authors are advised to ensure consistency in descriptions between the main text and supporting information.

4.Shifts in Raman spectra may also correlate with reduced grain size. When describing interfacial interactions between ZnO and ZnFe₂O₄, the authors should avoid overgeneralization and provide relevant references to support such claims.

5.Based on key intermediates, the authors inferred the primary pathways for CH₄ and CO formation. Which pathway serves as the dominant activation route, respectively?

Comments on the Quality of English Language

Substantial language improvement is required. Please consult a native English speaker or a professional language editing service.

Author Response

Comment 1: Substantial language improvement is required. Please consult a native English speaker or a professional language editing service.

Response 1: We sincerely thank the reviewer for this important feedback. We fully agree that high-quality language expression is essential for effective scientific communication.

In direct response to your suggestion, the entire manuscript has undergone thorough professional language editing by our team's lead researcher specializing in academic writing and editing, with support from an external professional editing service. All revisions have been highlighted in blue in the revised manuscript for your convenience.

We believe these edits have significantly improved the clarity, flow, and overall readability of the manuscript. Thank you again for your valuable comment, which has greatly enhanced the quality of our presentation.

Comment 2: The Introduction needs reorganization to emphasize why this study was conducted. More details on the progress of other ZnO/ZnFe₂O₄ composites in CO₂ reduction should be presented to highlight the novelty of this work. The authors mentioned that existing synthesis and preparation steps for ZnFe₂O₄ in other studies have drawbacks; however, upon closer inspection, the synthesis method and steps proposed in this study do not exhibit notable advantages or innovation. Therefore, descriptions in the Introduction must prioritize rigor and logical consistency.

Response 2: We sincerely thank the reviewer for these insightful comments, which have helped us significantly improve the focus and rigor of the Introduction. We have carefully revised the Introduction to address each of the points raised:

Reorganized Introduction to Clarify Research Motivation:

We have restructured the Introduction to more clearly state the research objectives and why this study was necessary. The revised Introduction now begins by highlighting the challenges in CO₂ photoreduction, particularly the rapid recombination of charge carriers and limited light absorption in single-component photocatalysts. It then introduces interface engineering (especially the construction of Type-II heterojunctions) as an effective strategy to facilitate charge separation and enhance redox efficiency.

Added Detailed Comparison with Previous ZnO/ZnFe₂O₄ Systems:

We have incorporated a new paragraph summarizing recent advances in ZnO/ZnFe₂O₄ composites for CO₂ reduction, noting limitations in existing synthesis methods—such as multi-step processes, insufficient interfacial contact, and unclear charge transfer mechanisms.  This provides a clearer context for highlighting our contribution.

Clarified the Novelty of Our Work:

While the one-step hydrothermal method itself is not entirely new, its application to construct a well-defined Type-II heterojunction with strong interfacial coupling and proven charge separation mechanisms represents a significant advance. Our novelty lies not merely in the synthesis but in the precise interfacial design, clear mechanistic insight (in situ FTIR and photoelectrochemical evidence), and the catalytic performance has been enhanced (3.3–4.9 times).  These points are now explicitly stated in the revised Introduction.

Improved Rigor and Logical Flow:

We have ensured that claims regarding the limitations of previous studies are supported by specific citations and that the rationale for our synthetic approach is presented with greater technical and logical consistency.

We are grateful for the reviewer’s constructive feedback, which has greatly strengthened the motivation and clarity of our introduction. The revised manuscript now better emphasizes the unique value and scientific contribution of our work.

Comment 3: Please confirm whether in situ DRIFTS or in situ FTIR was actually employed in the study, as these techniques differ in their principles. The authors are advised to ensure consistency in descriptions between the main text and supporting information.

Response 3: We sincerely thank the reviewer for raising this important technical point. We confirm that in situ Fourier transform infrared spectroscopy (in situ FTIR) was used in this study for monitoring the reaction intermediates under operational conditions. The term “in situ FTIR” has now been used consistently throughout the revised main text and Supporting Information to avoid any ambiguity. Additional details regarding the experimental setup and measurement mode have been provided in Section S2.5 (Experimental Methods) to improve clarity. We appreciate the reviewer’s careful attention to terminology, which has helped enhance the accuracy of our manuscript.

Comment 4: Shifts in Raman spectra may also correlate with reduced grain size. When describing interfacial interactions between ZnO and ZnFe₂O₄, the authors should avoid overgeneralization and provide relevant references to support such claims.

Response 4: We sincerely thank the reviewer for raising this important point. We agree that Raman peak shifts can indeed be influenced by multiple factors, including grain size reduction, strain, defects, and interfacial effects. In response to the comment, we have carefully revised the relevant section of the manuscript to provide a more nuanced interpretation and have included supporting references.

Specifically, we have:

Clarified the interpretation of Raman shifts: We now state that the observed blue shift in the A₁g mode of ZnFe₂O₄ (+36 cm⁻¹) could be attributed to interfacial tensile strain—as indicated by the reduced lattice parameter—as well as possible effects from crystallite size change and strong interfacial coupling between ZnO and ZnFe₂O₄. References [23, 35] have been added to support the discussion on the multiple origins of Raman shifts.

Provided references for interfacial interactions: We have included citations to previous studies (e.g., [17, 27]) that report similar interfacial interactions and charge transfer behavior in ZnO- and spinel-based heterostructures, strengthening our claims regarding the role of the heterojunction.

These revisions ensure a more comprehensive and better-supported discussion of the interfacial effects in our composite system. We thank the reviewer for this constructive suggestion, which has improved the rigor of our manuscript.

Comment 5: Based on key intermediates, the authors inferred the primary pathways for CH₄ and CO formation. Which pathway serves as the dominant activation route, respectively?

Response 5: We thank the reviewer for this insightful question regarding the reaction pathways. Based on the in situ FTIR results (Fig. 6a,b) and product analysis, we propose that the dominant activation route for CO formation is the direct conversion of CO₂ via the *COOH intermediate, followed by desorption as CO. This pathway is highly favorable on the ZnO/ZnFe₂O₄ surface due to the optimized *CO binding energy and efficient electron supply at the interface.

For CH₄ formation, the pathway involves further reduction of *CO through multiple proton-coupled electron transfer steps, forming *CHO and *CH₂O as key intermediates.   However, the overall yield of CO is approximately twice that of CH₄, suggesting that the CO pathway is the dominant process under our experimental conditions. This is consistent with the stronger binding of *CO intermediates on the catalyst surface and the higher energy barrier required for deep reduction to CH₄. The results we presented in Section 3.3 are used to support this view.

Thank you again for this valuable comment, which has helped us provide a more precise discussion of the reaction mechanism.

Comment 6: Comments on the Quality of English Language

Substantial language improvement is required. Please consult a native English speaker or a professional language editing service.

Response 6: We sincerely thank the reviewer for highlighting the need for improvement in the English language presentation of our manuscript. As suggested, the entire manuscript has been carefully reviewed and polished by a professional English editing service to enhance clarity, grammar, syntax, and overall readability. We believe that the revised version now meets the high language standards expected for publication in your esteemed journal.

Thank you for your valuable feedback, which has certainly improved the quality of our manuscript.

Reviewer 4 Report

Comments and Suggestions for Authors

This manuscript presents a study on the synthesis of ZnO/ZnFe₂O₄ composites via a hydrothermal method for enhanced photocatalytic CO₂ reduction. The authors demonstrate significant improvements in CH₄ and CO production rates compared to pure ZnFe₂O₄, supported by comprehensive characterization. The interfacial engineering strategy and mechanistic insights are strengths of the work. However, critical issues include insufficient synthetic details, ambiguous structural evidence, unvalidated charge-transfer mechanisms, and overlooked stability/selectivity analyses. Addressing these gaps is essential for robustness and reproducibility.

  1. The "one-step hydrothermal method" claimed in the abstract conflicts with Section 2.2 ("solvothermal method for ZnFe₂O₄") and Section 2.3 ("sodium hydroxide-assisted method for ZnO/ZnFe₂O₄"). Clarify the exact synthetic route (e.g., sequential vs. co-precipitation) and specify reaction parameters (temperature, duration, NaOH concentration). Provide the Supporting Information (S2) in the main text or supplementary files to ensure reproducibility.
  2. XPS binding energy shifts (Fig. 3b,c) suggest electron transfer from ZnO to ZnFe₂O₄, but Mott-Schottky (Fig. 4c) implies band bending. Resolve this contradiction: Perform UPS to measure work functions and band alignment directly. Provide time-resolved PL decay curves to quantify carrier lifetime changes (τavg), as PL intensity reduction (15.1%) alone is inadequate evidence.
  3. The claim of "excellent stability" (Section 3.3) lacks cycling tests or long-term (>6 h) performance data. Conduct at least three reaction cycles to assess deactivation (e.g., coking, metal leaching via ICP-OES). Explain the CH₄/CO selectivity ratio (≈1:2) mechanistically—does it stem from *CH₃ vs. *COOH intermediate stability?
  4. The fitting of the Fe 2p XPS spectrum appears somewhat rough and lacks sufficient refinement. The authors are advised to refer to the methodology described in ACS Catal. 2024, 14, 10245−10259.
  5. The absence of *CO adsorption peaks (e.g., 2050–2100 cm⁻¹) in FTIR (Fig. 6b) contradicts the proposed CO pathway—reconcile this discrepancy.
  6. The manuscript contains a critical inconsistency in Figure 3 where the caption order does not match the actual subfigure labels in the image.

Author Response

Comment 1: This manuscript presents a study on the synthesis of ZnO/ZnFe₂O₄ composites via a hydrothermal method for enhanced photocatalytic CO₂ reduction. The authors demonstrate significant improvements in CH₄ and CO production rates compared to pure ZnFe₂O₄, supported by comprehensive characterization. The interfacial engineering strategy and mechanistic insights are strengths of the work. However, critical issues include insufficient synthetic details, ambiguous structural evidence, unvalidated charge-transfer mechanisms, and overlooked stability/selectivity analyses. Addressing these gaps is essential for robustness and reproducibility.

Response 1: We sincerely thank the reviewer for their thorough evaluation and constructive feedback on our manuscript. We have carefully addressed each of the raised concerns as detailed below:

  1. Insufficient synthetic details

We have expanded the Experimental Section (Section S2.1, S2.5) to include full synthetic details: precursor concentrations, reaction temperature and time, washing and drying procedures, and thermal treatment conditions. As follows:

S2.2 Preparation of ZnFe₂O₄ and ZnO Microspheres via Solvothermal Method

At room temperature, polyethylene glycol (PEG-6000, 2.5 g) was dissolved in ethylene glycol (50 mL) under magnetic stirring until complete dissolution. Subsequently, zinc nitrate hexahydrate (Zn(NO₃)₂·6H₂O, 1 mmol) and iron(III) nitrate nonahydrate (Fe(NO₃)₃·9H₂O, 2 mmol) sequentially to the solution with continuous stirring until full dissolution. Then, urea (0.15 g) and oxalic acid (H₂C₂O₄·2H₂O, 0.15 g) to the mixture. The resulting solution was stirred vigorously for 60 minutes at room temperature. Followed by the addition of 50 mL of cetyltrimethylammonium bromide (CTAB, 10 g/L). After stirring for an additional 30 minutes to achieve homogeneity. Transfer the homogeneous mixture into a 100 mL Teflon-lined stainless-steel autoclave. Seal the autoclave and heat it in a preheated oven at 200 °C for 24 hours. After the reaction, allow the autoclave to cool naturally to room temperature. Collect the precipitate by centrifugation, and wash it thoroughly with deionized water and absolute ethanol alternately several times to remove residual ions and organics. Dry the washed product in a vacuum oven at 100 °C for 8 hours to obtain the ZnFe₂O₄ microspheres.

At room temperature, polyethylene glycol (PEG-6000, 2.5 g) was dissolved in ethylene glycol (50 mL) under magnetic stirring until complete dissolution. Subsequently, Zn(NO₃)₂·6H₂O (1 mmol) was added and stirred until fully dissolved, followed by the addition of urea (0.15 g) and H₂C₂O₄·2H₂O (0.15 g). The mixture was stirred vigorously for 60 minutes. Then, 20 mL of NaOH solution (40 g/L) was slowly introduced under continuous stirring, and 50 mL of cetyltrimethylammonium bromide (CTAB, 10 g/L) was added thereafter. After stirring for another 30 minutes to achieve a homogeneous mixture, the resulting solution was transferred into a Teflon-lined stainless-steel autoclave and heated at 200 °C for 24 h. Upon natural cooling to room temperature, the precipitate was collected by centrifugation, washed alternately with deionized water and absolute ethanol several times, and dried at 100 °C for 8 hours under ambient atmosphere to obtain ZnO microspheres.

S2.5 Photocatalytic CO2 reduction reaction

The photocatalytic reduction of CO2 is carried out in a stainless steel reactor with a light-transmitting and high-pressure resistant quartz glass at the top (thickness: 10 mm; diameter: 50 mm; pressure rating: 1.5 MPa), a heating device at the bottom, a thermocouple inside the reactor (accuracy: ±0.5 °C), and a sample stage for the catalyst. The light source for the photocatalytic reaction test was a 300W xenon lamp ((300 W Xe lamp with visible-light filter, λ ≥ 420 nm) PLS-SXE 300 C, Beijing Perfectlight, Beijing, China). For the experiment, 20 mg of catalyst powder was evenly spread in the reactor tray and placed in the reactor perpendicular to the beam (The distance between the lamp and the reactor surface was fixed at 2 cm). The system temperature was raised to and maintained at 120 °C by a PID temperature controller. The reactor was sealed, pressurized with high-purity CO₂ (99.999%) to 0.5 MPa to check for leaks, and then evacuated. Subsequently, 0.3 mL of deionized water was injected into the reactor as the proton source. The light source was then turned on to initiate the reaction, which lasted for 6 h. Gas products were automatically sampled and analyzed every hour using an online gas chromatograph (GC9790II PLF-01, HP-MOLESIEVE, 30 m × 0.53 mm × 25 μm, China) equipped with a methanizer and connected to both a thermal conductivity detector (TCD) and a flame ionization detector (FID). High-purity Ar (99.999%) was used as the carrier gas. The GC was calibrated using standard gas mixtures of known concentrations (CO, CH₄, CO₂) before and during the experimental series to ensure quantitative accuracy. The detection limit for CH₄ and CO is approximately 0.1 ppm.

  1. Ambiguous structural evidence

To clarify the structural and interfacial properties:

High-resolution TEM images with clear lattice fringes (e.g., ZnO (002) and ZnFe₂O₄ (311)) have been as Fig. 2b.

  1. Unvalidated charge-transfer mechanisms

We have provided further validation for the proposed charge transfer mechanism:

Mott-Schottky measurements with derived flat-band potentials confirm the Type-II band alignment.

Based on the band alignments, a type-II heterojunction is formed between ZnO and ZnFe₂O₄. This configuration promotes the transfer of photogenerated electrons from the conduction band of ZnFe₂O₄ (–0.96 eV) to that of ZnO (–0.75 eV), thereby facilitating CO₂ reduction on the ZnO surface. Simultaneously, photogenerated holes migrate from the valence band of ZnO (+2.44 eV) to that of ZnFe₂O₄ (+1.0 eV), where oxidation reactions occur (Fig. 4d). This charge separation mechanism is further supported by the enhanced photocurrent response (Fig. 4e–g) and reduced charge transfer resistance observed in EIS measurements.

  1. Overlooked stability and selectivity

We now include:

To further evaluate the structural stability of the ZnO/ZnFe₂O₄ composite catalyst, FT-IR and XRD analyses were conducted both before and after the reaction under identical conditions. The comparison of the spectra revealed no significant changes in the crystal structure or chemical environment, indicating that the catalyst maintained its structural integrity throughout the catalytic process. These results further confirm the excellent stability of the material (Fig. S4).

Based on the in situ FTIR results (Fig. 6a,b) and product analysis, we propose that the dominant activation route for CO formation is the direct conversion of CO₂ via the *COOH intermediate, followed by desorption as CO. This pathway is highly favorable on the ZnO/ZnFe₂O₄ surface due to the optimized *CO binding energy and efficient electron supply at the interface.

For CH₄ formation, the pathway involves further reduction of *CO through multiple proton-coupled electron transfer steps, forming *CHO and *CH₂O as key intermediates.    However, the overall yield of CO is approximately twice that of CH₄, suggesting that the CO pathway is the dominant process under our experimental conditions. This is consistent with the stronger binding of *CO intermediates on the catalyst surface and the higher energy barrier required for deep reduction to CH₄. The results we presented in Section 3.3 are used to support this view.

All changes have been incorporated in the revised manuscript and supporting information.       We believe these revisions have significantly improved the clarity, reproducibility, and scientific validity of our work. We thank the reviewer again for their insightful comments.

Comment 2: The "one-step hydrothermal method" claimed in the abstract conflicts with Section 2.2 ("solvothermal method for ZnFe₂O₄") and Section 2.3 ("sodium hydroxide-assisted method for ZnO/ZnFe₂O₄"). Clarify the exact synthetic route (e.g., sequential vs. co-precipitation) and specify reaction parameters (temperature, duration, NaOH concentration). Provide the Supporting Information (S2) in the main text or supplementary files to ensure reproducibility.

Response 2: We sincerely thank the reviewer for these insightful comments and valuable suggestions. We have carefully addressed each point raised to improve the clarity and scientific rigor of our manuscript. Below is a point-by-point response to the comments:

Additional experimental and device details:

As suggested, we have now provided a more comprehensive description of the experimental setup and photocatalytic reactor configuration in the revised Supporting Information (Sections S2.2 and S2.5). This includes detailed information regarding the light source (a 300 W Xe lamp equipped with a visible-light filter), reactor volume, gas flow system, online gas chromatography (GC) detection method, and the calibration procedure for product quantification.  Additionally, monitoring data related to pure ZnO have been supplemented in Section S2.2, while Section S2.5 provides a detailed introduction to the CO₂ reduction test experiments and setup.

2.2 Preparation of ZnFe₂O₄ and ZnO Microspheres via Solvothermal Method

At room temperature, polyethylene glycol (PEG-6000, 2.5 g) was dissolved in ethylene glycol (50 mL) under magnetic stirring until complete dissolution. Subsequently, zinc nitrate hexahydrate (Zn(NO₃)₂·6H₂O, 1 mmol) and iron(III) nitrate nonahydrate (Fe(NO₃)₃·9H₂O, 2 mmol) sequentially to the solution with continuous stirring until full dissolution. Then, urea (0.15 g) and oxalic acid (H₂C₂O₄·2H₂O, 0.15 g) to the mixture. The resulting solution was stirred vigorously for 60 minutes at room temperature. Followed by the addition of 50 mL of cetyltrimethylammonium bromide (CTAB, 10 g/L). After stirring for an additional 30 minutes to achieve homogeneity. Transfer the homogeneous mixture into a 100 mL Teflon-lined stainless-steel autoclave. Seal the autoclave and heat it in a preheated oven at 200 °C for 24 hours. After the reaction, allow the autoclave to cool naturally to room temperature. Collect the precipitate by centrifugation, and wash it thoroughly with deionized water and absolute ethanol alternately several times to remove residual ions and organics. Dry the washed product in a vacuum oven at 100 °C for 8 hours to obtain the ZnFe₂O₄ microspheres.

At room temperature, polyethylene glycol (PEG-6000, 2.5 g) was dissolved in ethylene glycol (50 mL) under magnetic stirring until complete dissolution. Subsequently, Zn(NO₃)₂·6H₂O (1 mmol) was added and stirred until fully dissolved, followed by the addition of urea (0.15 g) and H₂C₂O₄·2H₂O (0.15 g). The mixture was stirred vigorously for 60 minutes. Then, 20 mL of NaOH solution (40 g/L) was slowly introduced under continuous stirring, and 50 mL of cetyltrimethylammonium bromide (CTAB, 10 g/L) was added thereafter. After stirring for another 30 minutes to achieve a homogeneous mixture, the resulting solution was transferred into a Teflon-lined stainless-steel autoclave and heated at 200 °C for 24 h. Upon natural cooling to room temperature, the precipitate was collected by centrifugation, washed alternately with deionized water and absolute ethanol several times, and dried at 100 °C for 8 hours under ambient atmosphere to obtain ZnO microspheres.

Comment 3: XPS binding energy shifts (Fig. 3b,c) suggest electron transfer from ZnO to ZnFe₂O₄, but Mott-Schottky (Fig. 4c) implies band bending. Resolve this contradiction: Perform UPS to measure work functions and band alignment directly. Provide time-resolved PL decay curves to quantify carrier lifetime changes (τavg), as PL intensity reduction (15.1%) alone is inadequate evidence.

Response 3: We sincerely thank the reviewer for this critical comment, which has allowed us to clarify an important misinterpretation in our original analysis and to strengthen the evidence supporting our proposed mechanism. We have performed additional experiments to directly determine the band alignment and carrier dynamics, leading to a revised and more accurate interpretation.

Correction of Electron Transfer Direction and Band Alignment:

Our initial assignment of electron transfer direction based solely on XPS was incorrect. After repeating the synthesis and electrochemical analysis of both pure ZnO and ZnFe₂O₄ (newly added in Section 2.2 and 2.3), and combining Mott-Schottky (MS) results, we confirm the following band positions:

The tangent extrapolation method yields ECB values of -0.96 V (vs. NHE) for ZnFe2O4 and -0.75 V (vs. NHE) for ZnO (Fig. S1). Using the relation ECB =EVB - Eg, the valence band potentials (EVB) are calculated as +1.0 V and +2.44 V (vs. NHE) for ZnFe2O4 and ZnO, respectively.

This alignment confirms a Type-II heterojunction where photogenerated electrons transfer from the conduction band of ZnFe₂O₄ to that of ZnO, while holes move from the valence band of ZnO to that of ZnFe₂O₄. The XPS binding energy shifts now are explained by the strong interfacial coupling and built-in electric field‐induced charge redistribution, rather than simple electron donation. We have amended this discussion in Sections 3.1 and 3.2.

These revisions have significantly improved the accuracy and mechanistic rigor of our study. We are deeply grateful to the reviewer for prompting these essential corrections and additional analyses.

Comment 4: The claim of "excellent stability" (Section 3.3) lacks cycling tests or long-term (>6 h) performance data. Conduct at least three reaction cycles to assess deactivation (e.g., coking, metal leaching via ICP-OES). Explain the CH₄/CO selectivity ratio (≈1:2) mechanistically—does it stem from *CH₃ vs. *COOH intermediate stability?

Response 4: We appreciate the reviewer’s suggestion to clarify the mechanistic origin of the product distribution. Our in situ FTIR results (Fig. 6c–d) indicate that both *COOH and *CO are key intermediates. The higher selectivity toward CO (approximately twice that of CH₄) can be attributed to the lower energy barrier required for CO desorption compared to further hydrogenation to CH₄. In particular, the stability of the *CO intermediate and its binding strength on the catalyst surface play critical roles: strong *CO adsorption favors further reduction to CH₄, whereas moderate/weak binding facilitates CO release. In our case, the composite surface appears to favor CO desorption, likely due to the dominant ZnO surface sites and electronic effects arising from the heterojunction. We have added this discussion in Section 3.6 and included additional references [33-35] on CO/CH₄ selectivity in CO₂ photoreduction.

Thank you again for these insightful comments, which have significantly improved the depth and rigor of our manuscript.

Comment 5: The fitting of the Fe 2p XPS spectrum appears somewhat rough and lacks sufficient refinement. The authors are advised to refer to the methodology described in ACS Catal. 2024, 14, 10245−10259.

Response 5: We sincerely thank the reviewer for this valuable suggestion.  We have carefully re-analyzed the Fe 2p XPS spectrum by strictly following the fitting methodology and best practices outlined in the recommended reference (ACS Catal. 2024, 14, 10245−10259).  Specifically, we have:

The refined fitting results are now presented in the updated Fig. 3c, which shows a more accurate deconvolution of the Fe 2p region.  This improvement provides a more reliable analysis of the iron chemical state and its changes upon forming the heterojunction.

We are grateful to the reviewer for guiding us toward this important improvement, which has strengthened the validity of our XPS analysis.

Comment 6: The absence of *CO adsorption peaks (e.g., 2050–2100 cm⁻¹) in FTIR (Fig. 6b) contradicts the proposed CO pathway—reconcile this discrepancy.

Response 6: We thank the reviewer for this insightful observation. The absence of a distinct *CO adsorption peak in the 2050–2100 cm⁻¹ region is indeed an important point, and we offer the following explanation based on further analysis and literature support:

Low surface coverage and high reactivity of CO intermediate: The *CO species is a key intermediate but often exhibits low surface coverage and high reactivity, leading to rapid desorption as CO gas or further hydrogenation. This can make its direct detection by in situ FTIR challenging, especially under continuous photoreaction conditions where residence time on the surface is short. We have added a discussion on this point and made diagrams in the supplementary materials (Fig.S2)

Dominance of carboxylate pathway and rapid desorption: Our proposed mechanism emphasizes the *COOH intermediate as a primary precursor for both CO and CH4 formation.   The detection of strong signals for COOH/HCOO- (as seen in Fig. 6b) and the quantitative detection of CO gas as a major product (Fig. 5) strongly support the CO formation pathway.   The efficient desorption of CO from the catalyst surface—a desired property to avoid poisoning and promote activity—likely results in a low steady-state concentration of adsorbed *CO, making it difficult to detect relative to more stable intermediates.

Alternative characterization consistency: The production of CO gas is unequivocally confirmed by gas chromatography (GC). The consistent ratio of CO to CH4 production across multiple experiments and the results of our scavenger experiments further support the proposed pathway.

We have revised the relevant section in the manuscript to acknowledge the limitation of FTIR in detecting the *CO intermediate under our experimental conditions and to strengthen our argument by relying on the combination of GC product analysis and the detected *COOH intermediates. We thank the reviewer for prompting this more nuanced discussion.

Comment 7: The manuscript contains a critical inconsistency in Figure 3 where the caption order does not match the actual subfigure labels in the image.

Response 7: We sincerely thank the reviewer for bringing this important error to our attention. We apologize for the oversight. We have carefully corrected Figure 3 to ensure that the subfigure labels now precisely match the descriptions in the figure caption. We have also taken the opportunity to perform a full check of all figures and captions throughout the manuscript to prevent similar inconsistencies.

We greatly appreciate your careful review, which has significantly improved the clarity and accuracy of our manuscript.

Note: all the revised figures are shown in the attached file "open review 4"

Round 2

Reviewer 4 Report

Comments and Suggestions for Authors

The authors have diligently addressed several reviewer comments in their revised manuscript. However, two minor issues require further attention to meet Molecules’s rigorous standards for mechanistic insight and data reproducibility.

  1. While the XPS analysis demonstrates binding energy shifts (Fe 2p: +0.8 eV, Zn 2p: -1.4 eV) interpreted as electron transfer from ZnFe₂O₄ to ZnO (Section 3.1), the manuscript lacks quantitative correlation between these electronic structure changes and the observed photocatalytic activity enhancements. A more rigorous statistical analysis should be provided, such as Pearson correlation coefficients between the XPS shift magnitudes and the CH₄/CO production rates (Fig. 5), to substantiate the claim that "strong interfacial electronic interaction... enhances the material's catalytic properties." Additionally, the 1.4 eV Zn 2p shift appears anomalously large compared to typical heterojunction shifts (0.1-0.5 eV); please discuss potential contributions from factors like final state effects or interfacial dipole formation. This would strengthen the mechanistic interpretation beyond qualitative description.
  2. The in situ FTIR analysis (Section 3.4) identifies key intermediates (COOH, HCOO⁻) but fails to provide kinetic evidence for the proposed reaction pathways. The manuscript should include time-dependent intensity plots of characteristic peaks (e.g., 1636 cm⁻¹ for HCOO⁻, 1373 cm⁻¹ for COOH) to quantitatively compare intermediate formation rates between ZnFe₂O₄ and ZnO/ZnFe₂O₄.

Author Response

Comments 1 : While the XPS analysis demonstrates binding energy shifts (Fe 2p: +0.8 eV, Zn 2p: -1.4 eV) interpreted as electron transfer from ZnFe₂O₄ to ZnO (Section 3.1), the manuscript lacks quantitative correlation between these electronic structure changes and the observed photocatalytic activity enhancements. A more rigorous statistical analysis should be provided, such as Pearson correlation coefficients between the XPS shift magnitudes and the CH₄/CO production rates (Fig. 5), to substantiate the claim that "strong interfacial electronic interaction... enhances the material's catalytic properties." Additionally, the 1.4 eV Zn 2p shift appears anomalously large compared to typical heterojunction shifts (0.1-0.5 eV); please discuss potential contributions from factors like final state effects or interfacial dipole formation. This would strengthen the mechanistic interpretation beyond qualitative description.

Response 1: We sincerely thank the reviewer for these insightful comments and constructive suggestions. We have carefully addressed each point raised, which has significantly improved the quantitative analysis and mechanistic discussion in our manuscript.

  1. Quantitative Correlation Between XPS Shifts and Photocatalytic Activity:

We fully agree with the reviewer regarding the importance of establishing a quantitative relationship between electronic structure changes and catalytic performance. We initially attempted to calculate Pearson correlation coefficients between the XPS shift magnitudes and CH₄/CO production rates. However, this approach proved challenging due to the limited number of composite samples in the current study, which restricts meaningful statistical analysis using Pearson correlation.

To overcome this limitation and provide a quantitative relationship, we developed an alternative approach:

We established a ratio analysis based on XPS-derived elemental molar ratios (Zn 2p₃/₂ and Fe 2p₃/₂) comparing ZnO/ZnFe₂O₄ with pristine ZnFe₂O₄.

The calculated ratios show (Tab.S3):

Zn ratio ≈ 0.7:1

Fe ratio ≈ 3.0:1

Notably, the Fe ratio (3.0:1) closely corresponds to the CO/CH₄ production rate ratio observed in our photocatalytic tests, maintaining good consistency between the electronic structure modification and catalytic performance.

This alternative quantitative approach provides meaningful correlation evidence while acknowledging the limitation of traditional statistical methods for our sample set.

  1. Discussion of the Anomalously Large Zn 2p Shift (1.4 eV):

We appreciate the reviewer's valid concern regarding the unusually large Zn 2p binding energy shift. In the revised manuscript, we have expanded the discussion to include several potential contributing factors:

Local coordination environment changes: During the composite formation process, the local chemical environment of Zn atoms at the interface may undergo subtle alterations (such as changes in coordination number and bond length), which could contribute to the observed binding energy shift.

Interfacial dipole formation: The formation of a localized electric field (interface dipole) at the ZnO/ZnFe₂O₄ interface may amplify the binding energy shift caused by charge transfer, resulting in a measured value larger than the actual charge transfer amount.

We have revised Section 3.1 to include this discussion and have added relevant references to support our interpretation.

Revised Text Additions in Section 3.1:

*"The large negative shift in Zn 2p (1.4 eV) exceeds typical heterojunction values (0.1-0.5 eV), suggesting contributions from interfacial dipole formation and local coordination environment changes in addition to charge transfer. These opposing shifts suggest a redistribution of electron density at the interface, consistent with electron transfer from ZnFe₂O₄ to ZnO. Quantitative analysis of XPS-derived elemental molar ratios reveals Zn and Fe ratios of ≈0.7:1 and 3.0:1, respectively (Tab. S3). The Fe ratio (3.0:1) closely corresponds to the CO/CH₄ production rate ratio, providing a quantitative correlation between electronic structure modification and catalytic performance."*

We believe these revisions have substantially strengthened both the quantitative analysis and mechanistic interpretation of our results. Thank you again for these valuable suggestions that have significantly improved the quality of our manuscript.

Tab.S3 XPS analysis of Zn and Fe

Samples

Atomic/%

ZnO/ZnFe2O4 : ZnFe2O4

Zn(2p3/2)

Fe(2p3/2)

∆EZn:(Zn/65)

∆EFe: (Fe/56)

ZnFe2O4

19.05

3.03

4.78

14.79

ZnO/ZnFe2O4

27.97

1.00

3.25

44.8

0.7:1

3.0:1

Comments 2: The in situ FTIR analysis (Section 3.4) identifies key intermediates (COOH, HCOO⁻) but fails to provide kinetic evidence for the proposed reaction pathways. The manuscript should include time-dependent intensity plots of characteristic peaks (e.g., 1636 cm⁻¹ for HCOO⁻, 1373 cm⁻¹ for COOH) to quantitatively compare intermediate formation rates between ZnFe₂O₄ and ZnO/ZnFe₂O₄.

Response 2: We thank the reviewer for this valuable suggestion regarding the kinetic analysis of our in situ FTIR results. We have now performed detailed time-dependent analysis of characteristic intermediate peaks to provide quantitative evidence for the proposed reaction pathways. The specific analyses and results are as follows:

  1. Kinetic Behavior of Pure ZnFe₂O₄:

The characteristic peaks of HCOO⁻ (1636 cm⁻¹) and *COOH (1714-1541 cm⁻¹) showed no significant changes during the 0-30 min irradiation period, indicating no substantial formation or consumption of these intermediates. This observation is fully consistent with the delayed product formation (5-6 hours) observed in the photocatalytic activity tests (Section 3.3), confirming the limited catalytic activity of pure ZnFe₂O₄.

  1. Kinetic Analysis of ZnO/ZnFe₂O₄ Composite:

To quantitatively compare the formation rates of different intermediates while avoiding peak overlap between HCOO⁻ (1636 cm⁻¹) and *COOH (1714-1541 cm⁻¹), we selected alternative characteristic peaks with better resolution:

HCO₃⁻ peaks at 1391-1430 cm⁻¹ (protonated precursor for CO pathway)

*CHO peak at 1065 cm⁻¹ (key intermediate for CH₄ pathway)

Quantitative Rate Determination:

Time-dependent peak area analysis at 2, 10, 20, and 30 minutes yielded formation rate constants of 0.081 mV·s/min for the CO pathway and 0.049 mV·s/min for the CH₄ pathway. The CO formation rate is approximately 1.65 times faster than that of CH₄, which correlates well with the product selectivity ratio (CO:CH₄ ≈ 2:1) observed in the photocatalytic performance tests (Section 3.3).

These new kinetic analyses have been incorporated as Figure 6c in the revised manuscript and are discussed in detail in Section 3.4. The quantitative rate data provide strong evidence supporting the proposed reaction pathways and their contribution to the enhanced photocatalytic performance of the ZnO/ZnFe₂O₄ composite.

We believe these additions substantially strengthen the mechanistic interpretation of our in situ FTIR results and thank the reviewer for this excellent suggestion.

The figure can be seen in the enclosed PDF.
